# Hybrid Batch Normalisation: Resolving the Dilemma of Batch Normalisation in Federated Learning

Hongyao Chen [1]   Tianyang Xu[✉ 1]   Xiao-jun Wu [1]   Josef Kittler [2]

## Abstract

Batch Normalisation (BN) is widely used in conventional deep neural network training to harmonise the input-output distributions for each batch of data. However, federated learning, a distributed learning paradigm, faces the challenge of dealing with non-independent and identically distributed data among the client nodes. Due to the lack of a coherent methodology for updating BN statistical parameters, standard BN degrades the federated learning performance. To this end, it is urgent to explore an alternative normalisation solution for federated learning. In this work, we resolve the dilemma of the BN layer in federated learning by developing a customised normalisation approach, Hybrid Batch Normalisation (HBN). HBN separates the update of statistical parameters (*i.e.*, means and variances used for evaluation) from that of learnable parameters (*i.e.*, parameters that require gradient updates), obtaining unbiased estimates of global statistical parameters in distributed scenarios. In contrast with the existing solutions, we emphasise the supportive power of global statistics for federated learning. The HBN layer introduces a learnable hybrid distribution factor, allowing each computing node to adaptively mix the statistical parameters of the current batch with the global statistics. Our HBN can serve as a powerful plugin to advance federated learning performance. It reflects promising merits across a wide range of federated learning settings, especially for small batch sizes and heterogeneous data. Code is available at https://github.com/Hongyao-Chen/HybridBN.

[1]School of Artificial Intelligence and Computer Science, Jiangnan University, Wuxi, China. [2]School of Computer Science and Electronic Engineering and the Centre for Vision, Speech and Signal Processing (CVSSP), University of Surrey, Guildford, GU2 7XH, UK.. Correspondence to: Tianyang Xu <tianyang.xu@jiangnan.edu.cn>.

*Proceedings of the $42^{nd}$ International Conference on Machine Learning*, Vancouver, Canada. PMLR 267, 2025. Copyright 2025 by the author(s).

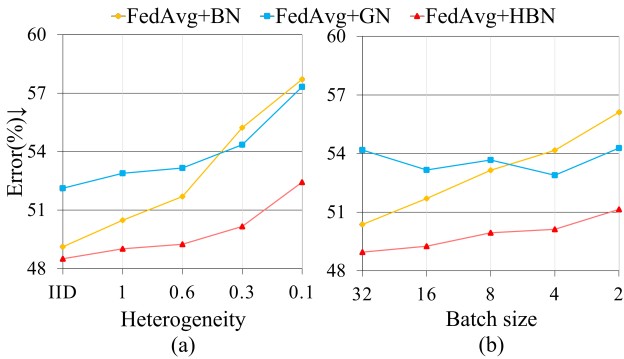

Figure 1: A comparison of our Hybrid Batch Normalisation (HBN) with standard Batch Normalisation (BN) and Group Normalisation (GN) in the federated learning settings. (a) **Classification error rate of Simple-CNN on CIFAR-100 *vs* Data heterogeneity** (controlled by a Dirichlet distribution coefficient). The batch size is 16. (b) **Classification error rate of Simple-CNN on CIFAR-100 *vs* Batch size.** The Dirichlet distribution coefficient is 0.6. For other implementation details, please refer to Section 4.1.

## 1. Introduction

Federated learning (FL) is a distributed machine learning paradigm that trains a global model on a central server, while protecting the raw data of each client node (Yang et al., 2019; Bharati et al., 2022). Different from data-centralised training, in FL, data across client nodes may be non-independently and identically distributed (Non-IID) (Zhao et al., 2018; Li et al., 2019; 2022), which has sparked a widespread interest in how to achieve the system-centralised performance during distributed training.

To guarantee data privacy, only the model parameters, rather than data, are allowed to be transferred between the central server and local clients. Specifically, the classic Federated Averaging (FedAvg) family (McMahan et al., 2017) performs multiple mini-batch stochastic gradient descent (SGD) on several local clients, and then aggregates the model parameters on the server in each communication round.

To harmonise the use of global model parameters in con-

junction with client specific data distributions, a suitable Batch Normalisation (BN) (Ioffe, 2015) is required. In the case of centralised training, samples from different batches are rescaled and shifted to a predefined range by the BN layer, reducing potential supervision conflicts. In general, the function of BN is to accelerate convergence (Bjorck et al., 2018; Karakida et al., 2019), to smooth the optimisation landscape (Santurkar et al., 2018; Yong et al., 2020; Peng et al., 2023), and to improve generalisation (Luo et al., 2018; Lubana et al., 2021), etc. However, (Hsieh et al., 2020) and (Du et al., 2022) found that the BN layer does not maintain its expected merits in the FL scheme. The underlying reason is that each local client works with a specific data distribution, resulting in heterogeneity from the global perspective. The difference between the local statistics (for training) pertaining to each client and the global statistics (for evaluation) compromises the effectiveness of the standard BN layer.

To effectively avoid collecting the statistics from each batch in FL, (Hsieh et al., 2020) suggests replacing BN with Group Normalisation (GN) (Wu & He, 2018), which performs a normalisation along the channel dimension. (Du et al., 2022) suggests using LN (Ba, 2016). Besides, (Kim et al., 2023; Zhang et al., 2024) identifies that Feature Normalisation(FN) can be helpful for federated training. Similar to GN, FN and LN both use the statistical information of a single sample for normalisation, ignoring the dependency relationships among samples. Although these normalisation methods avoid batch dependence by relying on individual samples, they fail to incorporate global information, which may further exacerbate data distribution heterogeneity.

Many studies (Ioffe, 2017; Yao et al., 2021; Pham et al., 2022) have found that the degree of heterogeneity and the batch size are the two dominant factors causing the global/local discrepancy for BN. As shown in Figure 1, when the heterogeneity of the data distribution is severe (with a smaller Dirichlet distribution coefficient) or the batch size is tiny, BN deteriorates because of the lack of confidence in the local statistics. In the decentralised training scenario, global statistical information is crucial for ensuring local consistency and synchronisation. At this point, normalisation based on global statistics is more appropriate. However, obtaining real-time global statistics is difficult in FL. Drawing on this, FedTAN (Wang et al., 2023) proposes to correct the potentially unreliable statistical parameters of BN. However, FedTAN requires more client-server communications to rectify the statistics, sacrificing $\times 3$ times as many BN layers in the communication rounds, compared to FedAvg.

To date, the potential of BN in handling FL tasks has not been sufficiently investigated. One challenging issue is how to gauge the client distribution heterogeneity, in relation to

the global statistical parameters. While the use of historical global statistical parameters for BN in decentralised training is precarious, we suggest using them to recognise heterogeneous nodes instead. Our approach demonstrates that this idea leads to an efficient solution for adapting the global statistical parameters of BN, so as to mitigate the negative impact of the distribution heterogeneity.

Recognising that the negative impact is produced during the update stage of statistics, where the local statistical parameters and learnable parameters are jointly updated, we propose to separate the update stage into sequential steps, *i.e.*, collecting the local statistical parameters, followed by updating the learnable parameters. In this way, our sequential update strategy recovers unbiased estimates of the last round's global statistics, without explicitly changing the communication transmission process. In the subsequent normalisation stage, we perform Hybrid Batch Normalisation (HBN) to balance the global statistical parameters and the statistical parameters from the current batch. In Figure 1, we compare HBN with standard BN and GN configured with the FedAvg baseline. Intuitively, HBN outperforms its contenders regardless of the actual degree of heterogeneity and batch size, achieving consistent normalisation for federated learning in diverse scenarios. We demonstrate that the proposed HBN can be used effectively in conjunction with several federated learning frameworks, resolving the dilemma of batch normalisation, and delivering consistently outstanding performance.

The main contributions of our work are as follows:

- We address the dilemma of batch normalisation for the federated learning framework, and show how to derive unbiased global statistical parameters from inconsistent local distributions.

- We develop a hybrid batch normalisation (HBN) layer, harmonising local batch statistics and global statistical parameters for each local client.

- We demonstrate the superiority of HBN by extensive experimental comparisons with several existing FL normalisation solutions. The experimental results show that HBN exhibits excellent merits under heterogeneous and small batch sizes scenarios.

- We verify the consistent compatibility of HBN with existing federated learning schemes and classic deep neural network architectures.

## 2. Related Work

**Learning Model Parameters in Federated Setting.** Compared to traditional distributed training, data heterogeneity among various local clients is a fundamental challenge in

federated learning (Kairouz et al., 2021; Vahidian et al., 2023; Mendieta et al., 2022). Due to privacy constraints, local clients cannot address this challenge by sharing data.

Since FedAvg (McMahan et al., 2017) was proposed to aggregate model parameters from multiple local clients, setting up a baseline for federated learning, recent studies typically focus on handling heterogeneous client data distributions. For instance, to alleviate oscillated aggregation at the central server end across different iterations, FedAvgM (Hsu et al., 2019) proposes to introduce momentum in the aggregation stage, aligning the entire model parameters. To achieve adaptive moment estimation (Kingma, 2014), FedAdam (Reddi et al., 2020) was developed to assign a suitable combination ratio between the previous and current parameters. To stabilise the local training phase, FedProx (Li et al., 2020) adds a proximal regularization term to local models. In terms of explicit anti-heterogeneous modelling, Scaffold (Karimireddy et al., 2020) reduces the variance of local updates by introducing a correction term for local gradients. In order to further collectorate the local and global models, FedCM (Xu et al., 2021) and FedSAM (Qu et al., 2022) introduce client-level momentum and client-level optimiser to enhance the generalisation power at the local end.

The above solutions harmonise the server-client parameters from the updating perspective of learnable parameters, achieving promising performance against heterogeneous distributions. On the contrary, in this paper, we focus on alleviating the negative impact of heterogeneity by designing a hybrid batch normalisation module, emphasising the update of statistical parameters, which can further boost the performance accompanied by the above solutions parallelly.

**Updating Statistical Parameters in Federated Setting.** Normalisation based on input statistics has been widely studied in deep learning, *e.g.*, batch normalisation, layer normalisation, and group normalisation. However, endowing federated learning solutions with the above normalisation approaches cannot deliver the expected improvement, remaining a concern on how to collect and update the statistical parameters in federated learning. For instance, batch normalisation relies on batch statistics to normalise data, which can be easily misled by the Non-IID clients' data (Ioffe, 2017). Drawing on this, Group Normalisation and Layer Normalisation are used for federated learning, avoiding collecting statistics across the batch dimension. Feature Normalisation alleviates the representation norm differences between federated learning clients, which is functionally similar to LN. However, GN, LN, and FN rely only on the statistics of each individual sample, representing significant difference compared to global statistics, which cannot maintain their advantages in heterogeneous FL. To preserve the statistical stability at the client end, FedBN (Li et al., 2021b) and

SiloBN (Andreux et al., 2020) limit certain BN parameters on each local client. The above approaches avoid relying on server-side statistical parameters and focus exclusively on local client normalisation, which are suitable for personalised federated learning but incapable of obtaining a global model.

To achieve server-client statistical parameters interaction and aggregation within the batch scope, recent studies have explored how to optimise BN in the FL setting. FixBN (Zhong et al., 2023), exploiting a straightforward strategy, proposes a two-stage training approach, where the statistical parameters are updated in the first stage and frozen to replace batch statistics during normalisation in the second stage. To involve historical statistics, Federated Batch Normalisation (FBN) (Guerraoui et al., 2024) proposes to normalise batch data using running average. However, FixBN and FBN neglect the diversity of statistics in each local client, which is a challenging issue in Non-IID scenarios. To simultaneously maintain the local data characteristics, FedTAN (Wang et al., 2023) uploads the local means and variances layer by layer to the server. Then, the global statistical parameters are sent to each client to update the local learnable parameters. Despite the improved accuracy in collecting statistics, FedTAN sacrifices $3\times$ times the communication cost for each BN layer compared to FedAvg.

Although existing attempts optimise the normalisation algorithms for FL, the potential value of BN statistics is not sufficiently explored during training. On the contrary, in this paper, we propose to harmonise the BN statistics with an unbiased estimator at the global server, which can endow a global perspective to each local training phase, unifying the task supervision thereby.

## 3. Approach

### 3.1. Formulation

**Batch Normalisation.** Batch Normalisation is widely used in deep neural networks to reduce internal covariate shift (Ioffe, 2015). Given an input feature $\{x_i\}_{i=1}^{B}$ of batch size $B$, BN calculates the means $\mu_{\text{b}}$ and variances $\sigma_{\text{b}}^2$ of the current batch and normalises them.

$$\mu_{\text{b}} = \frac{1}{B}\sum_{i=1}^{B} x_i \; ; \; \sigma_{\text{b}}^2 = \frac{1}{B}\sum_{i=1}^{B}(x_i - \mu_{\text{b}})^2. \qquad (1)$$

In general, BN estimates global statistical parameters for evaluation by employing the exponential moving average to update the running statistics.

In the current deep learning community, the estimated statistical parameters from the training phase are frozen in the evaluation phase. The vitality of these statistical parameters

holds only if the data share a similar distribution. Intuitively, independent and identically distributed (IID) data is expected. However, this assumption always conflicts with the federated learning setting, where heterogeneous data distributions are assigned to different local clients.

**Federated Learning with Batch Normalisation.** In the context of federated learning, handling non-IID data among local clients is a typical challenging issue. A direct solution is passing data, which is forbidden in the federated learning setting. Therefore, other local clients can access only model parameters to protect data privacy. Taking the classic FedAvg as an example, the goal of FL is to learn a global model that can perform well on all clients. Specifically, assuming that set $\mathcal{D}_g = \{\mathcal{D}_k\}_{k \in [K]}$ contains the datasets distributed across all $K$ clients. Considering a deep neural network with the BN layer, the model parameters include learnable parameters $\omega$ and BN's statistical parameters, which are means $\mu$ and variances $\sigma^2$. FL aims to minimise the following empirical risk:

$$\min_{\omega, \mu, \sigma^2} \mathcal{L}(\omega, \mu, \sigma^2) = \sum_{k=1}^{K} \frac{N_k}{N} \mathcal{L}_k(\omega, \mu, \sigma^2), \quad (2)$$

where $N_k$ is the size of the $k$-th client dataset $\mathcal{D}_k$, $N$ is the total size of data and $\mathcal{L}_k(\cdot)$ is the empirical risk of the $k$-th client. However, the above Equation (2) cannot be directly optimised because the central server cannot access the client data. For this purpose, FL iteratively aggregates the model parameters trained locally by the clients. On both global learnable parameters $\omega_g$ and global statistical parameters $\mu_g, \sigma_g^2$, FedAvg performs the following operations:

$$\{\omega_g, \mu_g, \sigma_g^2\} = \sum_{k=1}^{K} \frac{N_k}{N} \{\omega_k, \mu_k, \sigma_k^2\}, \quad (3)$$

where $\omega_k, \mu_k$ and $\sigma_k^2$ are all updated by the $k$-th client on local data.

**The Dilemma of Batch Normalisation.** Next, we focus on analysing the statistical parameters $\mu, \sigma^2$. We use

$$\mu, \sigma^2 = \mathcal{S}(\omega; \mathcal{D}), \quad (4)$$

to represent the statistical parameters $\mu, \sigma^2$ obtained by the model parameters $\omega$ on input data $\mathcal{D}$.

The dilemma of BN layer in federated learning mainly lies in our expectation of obtaining ideal global statistical parameters $\hat{\mu}_g, \hat{\sigma}_g^2$ during the training phase, which can be used for evaluation, *i.e.*,

$$\hat{\mu}_g, \hat{\sigma}_g^2 = \mathcal{S}(\omega_g; \mathcal{D}_g). \quad (5)$$

Unfortunately, previous methods approximate this process by averaging the local statistical parameters, *i.e.*,

$$\mu_g, \sigma_g^2 = \sum_{k=1}^{K} \frac{N_k}{N} \mathcal{S}(\omega_k; \mathcal{D}_k) \neq \mathcal{S}(\omega_g; \mathcal{D}_g). \quad (6)$$

---

**Algorithm 1** Workflow of Federated Learning with HBN

**Input:** the number of communication rounds $T$; the number of clients $K$; the datasets of clients $\{\mathcal{D}_k\}_{k \in [K]}$; the datasize of clients $\{N_k\}_{k \in [K]}$; the initial global model parameters $\{\omega_g, \mu_g, \sigma_g^2\}^0$ for the server; the initial hybrid distribution factor $\{\alpha_k = 0\}^0$ for each client $k$;

**for** $t = 1 \to T + 1$ **do**
  **for** each client $k$ in parallel **do**
    download $\{\omega_g, \mu_g, \sigma_g^2\}^{t-1}$ from server;
    // without backpropagation
    compute $\mu_k^t, (\sigma^2)_k^t$ according to Equation (7);
    // with backpropagation
    $\omega_k^t, \alpha_k^t \leftarrow \text{SGD}(\{\omega_g, \mu_g, \sigma_g^2, \alpha_k\}^{t-1}, \mathcal{D}_k)$;
    upload $\{\omega_k, \mu_k, \sigma_k^2\}^t$ to server;
  **end for**
  compute $\mu_g^t, (\sigma^2)_g^t$ according to Equation (8);
  **if** $t == T + 1$ **then**
    set $\omega_g^{T+1} = \omega_g^T$;
  **else**
    update $\omega_g^t = \sum_{k=1}^{K} \frac{N_k}{N} \omega_k^t$;
  **end if**
**end for**
**Output:** $\{\omega_g, \mu_g, \sigma_g^2\}^{T+1}$

---

This method obviously cannot accurately approximate ideal global statistical parameters. We provide a more detailed derivation in Appendix A.1. Firstly, the main reason is that the calculation of local statistical parameters is not based on the same global model, but on each local models, which would lead to external covariate shift (Du et al., 2022). Secondly, the direct average local variance may deviate from the actual global variance, and this error may accumulate during the forward propagation of deep neural networks. Furthermore, even if we restore the ideal statistical parameter estimation of BN, these statistical parameters cannot guarantee a positive impact on federated training. During local training, they still use batch data statistical parameters to normalise the input, losing awareness of global distribution. Driven by these issues, we propose a novel BN solution, Hybrid Batch Normalisation (HBN), for federated learning.

### 3.2. Global Server End: Obtaining Unbiased Statistical Parameters

To harmonise the distribution, obtaining an ideal global statistical estimation is a prerequisite for our hybrid batch normalisation. In order to ensure that the statistical parameters are not affected by the model (learnable parameters) transformation during each client training process, we propose an update strategy that separates the statistical parameters from the learnable parameters. As shown in Algorithm 1, after downloading the global model from the server, the local client calculates the statistical parameters of the local

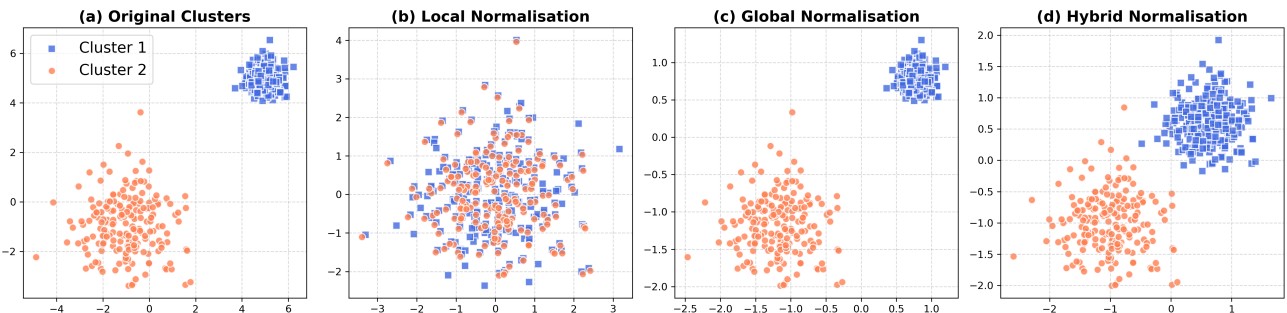

Figure 2: Normalisation methods for two toy FL clusters.

data based on the current global learnable parameters before starting training. So in the $t$-th round, client-$k$ can get:

$$\mu_k^t, (\sigma^2)_k^t = \mathcal{S}(\omega_g^{t-1}; \mathcal{D}_k). \qquad (7)$$

After obtaining the statistical parameters of the last global model, each client optimises the local learnable parameters based on standard stochastic gradient descent (SGD) with mini-batch. Usually, if local statistics are independent and randomly sampled from the entire data, then averaging them is reasonable, but in federated learning, this assumption does not hold. Drawing on this, we consider client data as a certain group of the entire population. According to distributed statistical analysis, we can use the following approach to calculate unbiased estimates of the population mean and variance:

$$\begin{cases} \mu_g^t = \sum_{k=1}^{K} \dfrac{N_k}{N} \mu_k^t; \\[2mm] (\sigma^2)_g^t = \sum_{k=1}^{K} \dfrac{N_k \left[ (\sigma^2)_k^t + (\mu_k^t - \mu_g^t)^2 \right]}{N - 1}. \end{cases} \qquad (8)$$

By separately calculating local statistical parameters, the server can obtain ideal global statistical parameters $\mathcal{S}(\omega_g^{t-1}; \mathcal{D}_g)$. We provide the detailed derivation in Appendix A.2. It should be noted that after using this asynchronous strategy to update between the statistical parameters and the learnable parameters, we need to perform an additional round of statistical parameters update for synchronisation at the end.

### 3.3. Local Client End: Performing Hybrid Batch Normalisation

In heterogeneous FL, local mini-batches sampled by each client reflect only their local data distribution, often differing substantially from the global distribution. As shown in Figure 2, two toy clusters follow Gaussian distributions, simulating the activation output of two clients in federated learning. Intuitively, after applying local normalisation to

each cluster separately using local statistics, the normalised data exhibits indistinguishable overlap compared to globally normalised data. Most local normalisation methods (BN, GN, LN, *etc.*) cannot produce satisfactory results in such cases. Compared to local normalisation, global normalisation can better maintain consistency among local clients. However, real-time global normalisation is impractical, particularly in federated learning, where models undergo multiple local updates before communication. This means the ideal global statistics are inherently dynamic. We usually only have access to historical global statistics, whose timeliness is constrained by intermittent communication and partial client participation. Using only historical global statistics will yield a suboptimal result. But historical global statistics still retain valuable global structural information. Especially since we have addressed the dilemma of global statistical parameter bias for BN, we can use these ready parameters more efficiently. Therefore, we propose a hybrid normalisation method that is more suitable for handling federated learning tasks, coined as hybrid batch normalisation (HBN). As shown in Figure 2 (d), this hybrid normalisation combines global statistics with local statistics, which can standardise the size of two clusters while maintaining the global structure.

Specifically, HBN adds a hybrid distribution factor $\alpha$ in the base BN layer to adaptively learn the balance between batch and global normalisation. Typically, the hybrid distribution factor $\alpha$ is a learnable parameter with the same size as the number of input channels. For a layer input $\{x_i\}_{i=1}^B$ over a mini-batch, we calculate the normalisation parameters for each input as follows:

$$\begin{aligned} \hat{\mu} &= \frac{e^{-\alpha}}{1 + e^{-\alpha}} \mu_b + \frac{1}{1 + e^{-\alpha}} \mu_g; \\ \hat{\sigma}^2 &= \frac{e^{-\alpha}}{1 + e^{-\alpha}} \sigma_b^2 + \frac{1}{1 + e^{-\alpha}} \sigma_g^2, \end{aligned} \qquad (9)$$

where $\mu_b, \sigma_b^2$ are the current batch statistics calculated according to Equation (1), and $\mu_g, \sigma_g^2$ are the global statistics stored in HBN according to Equation (8). In particular, the global statistic is frozen during the gradient update process.

Like other normalisation methods, HBN also learns a per-channel linear transform to compensate for the possible loss of representative power:

$$y_i = \gamma \frac{x_i - \hat{\mu}}{\sqrt{\hat{\sigma}^2 + \epsilon}} + \beta \equiv HBN_{\alpha,\gamma,\beta}(x_i), \qquad (10)$$

where $\epsilon$ is a small positive constant that prevents the denominator from being zero and $\gamma, \beta$ are trainable scale and offset.

Due to the varying degrees of distribution differences between clients and the global server, the hybrid distribution factor is configured for each local client, which does not participate in communication. To clarify this process, in Algorithm 1, we distinguish the hybrid distribution factor $\alpha$ from other learnable parameters $\omega$. In the training phase, the learnable parameters and local statistical parameters are updated separately and do not affect each other. In the evaluation phase, normalisation parameters use global statistical parameters $\{\mu_g, \sigma_g^2\}$ and do not require the participation of hybrid factor.

### 3.4. Implementation

**Comparison of parameters with BN.** For HBN, we can easily implement programming codes based on BN. HBN needs to store both local and global statistical parameters, but only one of them needs to participate in each communication round. Local statistics can be directly calculated on each client, while global statistics are obtained by aggregating local statistics on the server. In addition, we need to add trainable hybrid distribution factor parameters per channel and discard the moving average momentum. In the forward passing, HBN normalises the batch input in the order of Equations (1), (9) and (10).

**Communication with stragglers.** In situations where there are a large number of stragglers, inevitably, we are unable to calculate statistical measures for all data. We empirically recommend using the moving average to update the global statistical parameters for each server stage:

$$\mu_g^t = (1 - \lambda)\mu_g^{t-1} + \lambda \mu_g^t;$$
$$(\sigma^2)_g^t = (1 - \lambda)(\sigma^2)_g^{t-1} + \lambda(\sigma^2)_g^t, \qquad (11)$$

where $\lambda \in (0, 1]$ is the moving average momentum. For more stragglers, small momentum can provide more smooth estimates which is beneficial for training.

## 4. Experiments

### 4.1. Implementation Details

**Baselines.** Six normalisation baselines are considered, including Batch Normalisation (BN), Group Normalisation

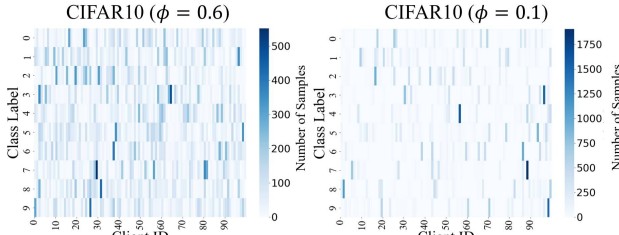

Figure 3: Different Dirichlet coefficients $\phi(0.6, 0.1)$ to label distribution on CIFAR-10 with 100 clients.

(GN), Layer Normalisation (LN), FedFN, FixBN, and Federated Batch Normalisation (FBN), For fairness, we default to combining all normalisation methods with the baseline FedAvg.

**Benchmark.** All experiments are conducted on the classic image classification datasets, including CIFAR-10/CIFAR-100 (Krizhevsky, 2009) and Tiny-ImageNet (100K images with 200 classes) (Chrabaszcz et al., 2017), which are widely adopted in FL research (Chen & Zhang, 2024). We use a customised Simple-CNN network on CIFAR-10/CIFAR-100, which has a representativeness in computer vision domain. Simple-CNN mainly consists of three convolutional blocks and two fully connected layers. Each convolutional block contains a convolutional layer, a normalisation layer, a ReLU activation layer, and a max-pooling layer (LeCun et al., 1998; Maas et al., 2013; Ioffe, 2015). We replace the normalisation layer with different normalisation methods, except for FN, which is added before the input of the classification head. We also use the classic ResNet-18 (He et al., 2016) on the large-scale dataset Tiny-ImageNet.

Similar to existing works (Wang et al., 2024; Tian et al., 2024; Zhuang et al., 2024), we use Dirichlet distribution (Hsu et al., 2019) to model the data distribution, which controls the degree of statistical heterogeneity through a coefficient $\phi$. The smaller the $\phi$, the more severe the label distribution heterogeneity across clients becomes. As shown in Figure 3, we visualise the label distribution of 100 clients on the CIFAR-10 with different $\phi(0.6, 0.1)$.

**Hyper-parameters.** For CIFAR-10/CIFAR-100/Tiny-ImageNet, unless otherwise specifie, we set the total number of clients $K = 100/100/500$, the activation rate of each round of clients $C = 0.1/0.1/0.02$ (*i.e.*, 10 clients participate in per round), the local training epoch $E = 1$, the optimiser is SGD optimiser with a momentum of 0.9. The communication rounds $T$ for CIFAR-10/CIFAR-100/Tiny-ImageNet are 500/1000/1000, respectively. For all algorithms, we select the appropriate initial learning rate from $\{0.02, 0.01, 0.005, 0.002, 0.001\}$ when batchsize is 4. The learning rates decay quadratically with 0.998. In particular, we adopt the linear learning rate scaling rule (Krizhevsky,

Table 1: Top-1 Test Accuracy (%) of different FL normalisation solutions on CIFAR-10/CIFAR-100/Tiny-ImageNet with batchsize 4/4/64.

| Setting | CIFAR-10 | | | CIFAR-100 | | | Tiny-ImageNet | |
|---|---|---|---|---|---|---|---|---|
| (FedAvg+) | $\phi = 0.6$ | $\phi = 0.3$ | $\phi = 0.1$ | $\phi = 0.6$ | $\phi = 0.3$ | $\phi = 0.1$ | $\phi = 0.1$ | $\phi = 0.05$ |
| BN (Ioffe, 2015) | 75.82 | 73.85 | 69.61 | 45.84 | 45.15 | 43.40 | 23.95 | 21.55 |
| GN (Wu & He, 2018) | 75.74 | 74.10 | 68.76 | 47.11 | 46.33 | 43.38 | 19.71 | 14.89 |
| LN (Ba, 2016) | 74.08 | 72.50 | 66.22 | 46.90 | 45.65 | 42.08 | 19.72 | 14.84 |
| FedFN (Kim et al., 2023) | 75.51 | 74.30 | 67.96 | 46.29 | 45.15 | 43.80 | 20.05 | 17.30 |
| FixBN (Zhong et al., 2023) | 77.34 | 74.51 | 70.36 | 46.05 | 45.54 | 42.15 | 24.71 | 23.42 |
| FBN (Guerraoui et al., 2024) | 73.91 | 71.94 | 66.13 | 45.35 | 43.72 | 41.61 | 4.90 | 3.76 |
| **HBN (ours)** | **78.22** | **76.53** | **71.56** | **49.88** | **48.93** | **46.52** | **25.59** | **24.69** |

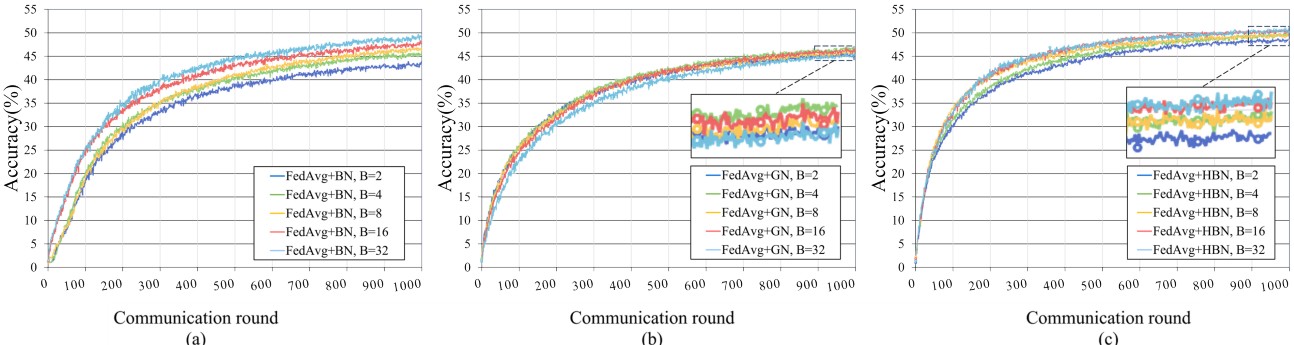

Figure 4: The sensitivity of different normalisation methods to batch size on CIFAR-100 with $\phi = 0.6$ by Simple-CNN.

Table 2: Test Accuracy (%) of different normalisation methods to batch size on CIFAR-100 with $\phi = 0.6/0.3$ by Simple-CNN.

| | $\phi = 0.6$ | | | $\phi = 0.3$ | | |
|---|---|---|---|---|---|---|
| | BN | GN | HBN | BN | GN | HBN |
| B=32 | 49.63 | 45.82 | **51.05** | 48.33 | 44.66 | **49.83** |
| B=16 | 48.30 | 46.84 | **50.75** | 49.52 | 45.65 | **49.84** |
| B=8 | 46.86 | 46.33 | **50.06** | 48.54 | 45.39 | **49.79** |
| B=4 | 45.84 | 47.11 | **49.88** | 45.15 | 46.33 | **48.93** |
| B=2 | 43.89 | 45.72 | **48.86** | 42.97 | 45.32 | **48.75** |
| Avg. | 46.90 | 46.36 | **50.12** | 46.90 | 45.46 | **49.42** |
| | ±2.21 | ±0.61 | **±0.85** | ±2.74 | ±0.60 | **±0.54** |

2014; Goyal et al., 2019) to adapt to batch size changes. For specific learning rate settings, please refer to the Appendix. The batch size $B$ and Dirichlet distribution coefficient $\phi$ will be specified in each experiment. For the moving average momentum in BN and FBN, we set it to 0.9. For GN, we set the number of groups to 2. For FixBN, we set the first half of the total number of rounds as the first stage and the rest as the second stage. For HBN, we set $\lambda$ to 0.01 in Equation (11).

Due to the mismatch between the statistical parameters and the learnable parameters of HBN in each communication round, we test the statistical parameters of the same client subset for accuracy evaluation.

### 4.2. Performance Comparison

**Influence of Data Heterogeneity.** Table 1 reports the test accuracy of all compared FL normalisation methods on CIFAR-10, CIFAR-100 and Tiny-ImageNet with various heterogeneous settings. We conduct experiments in two computing resource scenarios, Simple-CNN with batch size 4 on CIFAR-10/CIFAR-100 and ResNet-18 with batch size 64 on Tiny-ImageNet. Accordingly, although GN, LN and FN avoid dependence on global statistical parameters, they do not demonstrate outstanding performance in dealing with data heterogeneity. We find that traditional BN remains competitive in most scenarios. We believe that this is due to the moving average mechanism of BN, which prevents the locally obtained statistics from deviating too much from the population. FixBN and FBN employ shared global statistics for normalisation at different stages. When these global statistics are reliable, they benefit local training. Conversely, when the global statistics are unreliable, they exhibit detrimental effects. Our experimental scenario in Table 1 is deliberately challenging——featuring strong heterogeneity, numerous clients, restricted batchsize and frequent stragglers. In such conditions, FixBN and FBN fail to judge whether the local statistics, which are derived from diverse local models, can directly be aggregated, resulting in unreliable global statistics. Our proposed hybrid batch normalisation adaptively combines historical global statistics with real-time local statistics. Analogous to augmenting

Table 3: Experiments on CIFAR-10 across varying client numbers with $B = 4$ and $\phi = 0.6$ by Simple-CNN.

|       | K=100 | K=200 | K=500 | K=1000 |
|-------|-------|-------|-------|--------|
| BN    | 75.82 | 73.88 | 68.94 | 61.25  |
| GN    | 75.74 | 67.93 | 60.31 | 52.24  |
| LN    | 74.08 | 69.08 | 61.22 | 51.55  |
| FedFN | 75.51 | 70.71 | 62.02 | 53.75  |
| FixBN | 75.65 | 71.67 | 69.07 | 63.08  |
| FBN   | 73.91 | 68.21 | 59.22 | 50.73  |
| **HBN** | **78.22** | **75.76** | **72.49** | **64.95** |

local client training with global statistics, HBN dynamically integrates global information into the normalisation of current batches, achieving more effective normalisation.

**Sensitivity to Batch Sizes.** We collect the experimental results in terms of batch size in Table 2. We find that GN can alleviate the impact of batch size in federated learning as expected, but its generalisation power is worse than BN in large batch sizes. A larger batch size does not necessarily deliver better performance for GN, as it reduces the number of local updates. When the batch size is small, the performance of BN is greatly affected. For example, when the batch size is reduced from 32 to 2, the accuracy of BN drops by 5.74 percentage points, whereas HBN shows a smaller drop of only 2.19 percentage points. It is worth noting that even with a batch size of 2, the accuracy of HBN exceeds BN with a batch size of 32 and the optimal GN. Because HBN adaptively coordinate the global normalisation and the batch normalisation, HBN not only maintains the convergence speed of BN at appropriate batch sizes, but also reduces the sensitivity to small batch size. We provide a more intuitive visualisation in Figure 4.

**Different Number of Clients.** When the number of clients increases, methods based on local statistics are difficult to reflect the distribution of global data, and methods based on global statistics are difficult to obtain reliable global statistics under communication limitations. We conduct comparative experiments on CIFAR-10 ($\beta = 0.6$, $B = 4$) across varying client numbers (10 clients sampled per round). As shown in Table 3, HBN consistently outperforms baselines on all scales ($K = 100$ to $K = 1000$) through adaptive balance of these two factors.

**Consistent Boosting Power with Advanced FL Solutions.** As HBN belongs to the normalisation paradigm, it should act as a plug-in layer to other advanced FL solutions. We attempt to configure HBN on several advanced FL solutions, including FedProx, FedAdam, Moon, Scaffold, FedSAM, FedACG and Fedwon. FedAdam adds first-order and second-order moment estimates during global updates. FedProx and Scaffold alleviate local model bias by introducing additional constraint terms in local updates. Moon optimises local feature representation based on contrastive

Table 4: Test Accuracy (%) of different advanced FL solutions combined with HBN. The experiments are conducted on CIFAR-10 with $B = 4$ and $\phi = 0.6$ by Simple-CNN.

| Method | +BN/GN/- | +HBN |
|--------|----------|------|
| FedAvg (McMahan et al., 2017) | 75.82 | **78.22** |
| FedProx (Li et al., 2020) | 76.08 | **78.44** |
| FedAdam (Reddi et al., 2020) | 75.67 | **78.17** |
| Scafflod (Karimireddy et al., 2020) | 77.63 | **78.65** |
| Moon (Li et al., 2021a) | 75.77 | **78.48** |
| FedSAM (Qu et al., 2022) | 76.39 | **78.40** |
| FedACG (Kim et al., 2024) | 76.01 | **79.19** |
| Fedwon (Zhuang & Lyu, 2024) | 78.85 | **79.93** |

learning. FedSAM performs Sharpness Aware Minimisation local optimiser for local training. FedACG initialises local models using global momentum. Fedwon modifies the convolution layers directly to adjust the distribution, without using the standard normalisation layer. Because the update stage of FedAdam and FedACG conflicts with the statistical parameters in BN, resulting in a performance crash, we use GN for these two methods. We have not equipped any normalisation layer for Fedwon. For other methods, BN is used. Table 4 lists their basic test accuracy (+BN/GN/-) of Simple-CNN on CIFAR-10 with $B = 4$ and $\phi = 0.6$, as well as the test accuracy combined with HBN (+HBN). HBN obtains consistent gains in performance regardless of the federated learning approaches. The results demonstrate the potential of our HBN in delivering additional boosting power for the federated learning community.

Table 5: Test accuracy (%) of different model architectures with different normalisation layers on CIFAR-100 with $\phi = 0.1$ and $B = 64$.

| Model | +BN | +GN | +HBN |
|-------|-----|-----|------|
| MobilenetV2 | 40.19 | 41.50 | **44.40** |
| ResNet-18 | 51.51 | 52.38 | **55.69** |
| ResNet-50 | 54.97 | 55.37 | **60.74** |
| VGG-11 | 62.74 | 59.47 | **63.45** |
| VGG-19 | 64.09 | 61.39 | **64.56** |

**Consistent Compatibility with Classical Deep Networks Architectures.** In heterogeneous federated learning scenarios, deeper networks are more challenging in terms of training and convergence. Moreover, for vanilla BN statistical parameters, the variance obtained by directly weighted averaging is smaller than the true variance. For networks with more BN layers, this will result in more offset in the model output. We assemble HBN into deeper networks to verify its compatibility. We conduct experiments on CIFAR-100 with $\phi = 0.1$, adjusting the batch size to 64 while keeping all other settings the same as before. We mainly verify the compatibility of HBN in five pretrained network architectures, including MobilenetV2 (Sandler et al., 2018),

ResNet-18/50 (He et al., 2016) and VGG-11/19 (Simonyan & Zisserman, 2014). Usually, their original networks are assembled with BN. We replace BN with GN and HBN, and remove the dropout layer from the original network, which had no positive effect in our experiment. As shown in Table 5, HBN is suitable for most classic deep convolutional neural network architectures, achieving significant performance improvements in heterogeneous federated learning scenarios compared to BN and GN.

### 4.3. Ablation Study

**Component Analysis.** Compared with the vanilla BN, HBN modifies the operation in two aspects, calculating accurate statistical parameters (Equations (7) and (8)) and achieving adaptive normalisation (Equation (9)).

Table 6: Test Accuracy (%) of different configurations for HBN's each component. The experiments are conducted on CIFAR-10 with $B = 4$ and $\phi = 0.6$ by Simple-CNN.

| Index | Ablation Setting | Acc (%) |
|-------|------------------|---------|
| (1) | Equation (7) ← moving average | 72.25 |
| (2) | Equation (8) ← weighted average | 77.90 |
| (3) | Equation (9) ← only global statistics | 76.87 |
| (4) | Equation (9) ← only batch statistics | 76.33 |
| (5) | Equation (9) ← fix ratio at 1:1 | 77.49 |
| | FedAvg+HBN | **78.22** |

Table 6 displays the performance of using other candidates to replace each of our designed components. In principle, we freeze the other parts and only modify the following for each ablation experiment: (1) update local statistics based on moving average without separation, (2) summarise local statistics based on weighted average, (3) normalise only using global statistics, (4) normalise only using batch statistics, and (5) fix an equal hybridization ratio for global statistics and batch statistics.

We can see that replacing any component causes a negative impact on the performance of HBN. It is worth mentioning that in experiment (1), we do not separate the updates of the statistical parameters and the learnable parameters, which results in HBN being unable to benefit from unstable global statistical parameters and potentially exacerbating the shift of local statistics. In experiment (4), we only use batch statistics for normalisation, which is consistent with the vanilla BN. The improved performance indicates that separating the calculation of statistical parameters can compensate for the accuracy loss caused by the deviation of the vanilla BN statistical parameters.

**Selection of Hyper-parameters.** Figure 5 measures the impact of the choice of $\lambda$ in Equation (11) under different client activation ratios $C$. We select $C$ from [1, 0.7, 0.4, 0.1] and $\lambda$ from [1, 0.1, 0.01]. We conduct this experiment on

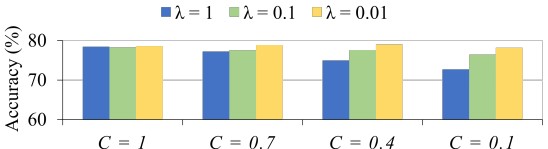

Figure 5: Impact of the hyper-parameter $\lambda$ under different client activation ratios $C$.

CIFAR-10 with $\phi = 0.6$, using Simple-CNN with a batch size of 32. We keep other settings consistent with those in Section 4.1. When the client activation rate is low, adding a moving average can alleviate the oscillation of global statistics and improve the performance of HBN. When the client activation rate is high, it is not sensitive to the selection of $\lambda$. According to our experiments, we recommend using a smaller $\lambda$, which usually performs satisfactorily.

## 5. Discussion

**Communication and Computation Overhead.** For federated learning, communication overhead is worth considering and is proportional to the parameter quantity. Compared to other methods, HBN only increases the communication cost for one round, which is negligible compared to the large number of communication rounds required for training. In terms of computational cost, HBN requires additional calculation of local statistics on the clients, but this part is much lower than the cost of model training. In addition, we find that using partial data to calculate local statistics on the client produces a marginally negative impact on model performance, as the client's local data usually follows the same distribution. For example, when we compute local statistics from randomly sampled 128 images per client, the final model performance decreases by only 0.13 percentage points.

## 6. Conclusion

In this work, we propose a targeted improvement for batch normalisation in federated learning, Hybrid Batch Normalisation. The Hybrid Batch Normalisation resolve the dilemma of batch normalisation statistical parameter mismatch in Federated Learning. In order to leverage the potential of global statistical parameters in solving Non-IID problems, Hybrid Batch Normalisation combines global normalisation and batch normalisation adaptively, alleviating vanilla batch normalisation limitations in small batch size and heterogeneous data. Extensive experiments validate the effectiveness of Hybrid Batch Normalisation. Our HBN extends the current batch normalisation scope in terms of the federated learning tasks.

## Acknowledgements

This work was supported in part by the National Key Research and Development Program of China under Grant No.2023YFE0116300, the National Natural Science Foundation of China (62020106012, 62332008, 62336004, 62106089), the Fundamental Research Funds for the Central Universities (JUSRP202504007).

## Impact Statement

This paper presents work whose goal is to advance the field of Machine Learning. There are many potential societal consequences of our work, none which we feel must be specifically highlighted here.

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

## A. Problem Formulation

### A.1. Statistical Bias of Vanilla BN in FL

In centralised training scenarios, internal covariate shift (Ioffe, 2015) refers to the change in the input distribution of each layer during the training of deep neural networks due to updates in model parameters. BN can effectively solve this problem under centralised training. However, in the context of federated learning, the changes in model parameters also occurs during the global aggregation stage. This results in the statistical measures obtained locally no longer being applicable globally, a phenomenon known as external covariate shift (Du et al., 2022). We can deduce that this statistical bias is caused by the heterogeneity of the distribution.

Let us consider the statistical bias of BN during one server-client communication caused by FedAvg. Assuming $\omega_g^{t-1}$ is the initial model of the server in round $t-1$. Each client $k \in [K]$ downloads the global model as the initialisation of the local model, and trains it on local dataset $\mathcal{D}_k$. Due to the fact that the statistical parameters in vanilla BN are updated synchronously with the learnable parameters, the statistical parameters obtained in each locality are approximate estimates of $\omega_k^t = \omega_g^{t-1} - \eta \nabla \mathcal{L}(\omega_g^{t-1}, \mathcal{D}_k)$ on $\mathcal{D}_k$, *i.e.*,

$$
\begin{aligned}
\mu_k^t &= \frac{1}{N_k} \sum_{i=1}^{N_k} f(\omega_k^t, x_k^i); \\
(\sigma^2)_k^t &= \frac{1}{N_k} \sum_{i=1}^{N_k} (f(\omega_k^t, x_k^i) - \mu_k^t)^2,
\end{aligned}
\tag{12}
$$

where $f(\cdot)$ is the output of model $\omega$ before normalising all layers of the sample $x$.

Then, the server aggregates the parameters of each client model, including the statistical parameters of BN. Ideally, we would like to obtain the following global statistics:

$$
\begin{aligned}
\hat{\mu}_g^t &= \frac{1}{N} \sum_{k=1}^{K} \sum_{i=1}^{N_k} f(\omega_g^t, x_k^i); \\
(\hat{\sigma}^2)_g^t &= \frac{1}{N-1} \sum_{k=1}^{K} \sum_{i=1}^{N_k} (f(\omega_g^t, x_k^i) - \hat{\mu}_g^t)^2,
\end{aligned}
\tag{13}
$$

where $\omega_g^t = \frac{N_k}{N} \sum_{k=1}^{K} \omega_k^t$.

However, there is a deviation between the average of local statistics and the ideal global statistics. We consider the difference between the mean $\mu_g^t = \frac{N_k}{N} \sum_{k=1}^{K} \mu_k^t$ and the ideal mean $\hat{\mu}_g^t$:

$$
\begin{aligned}
|\mu_g^t - \hat{\mu}_g^t| &= |\frac{N_k}{N} \sum_{k=1}^{K} \mu_k^t - \hat{\mu}_g^t| \\
&= |\frac{1}{N} \sum_{k=1}^{K} \sum_{i=1}^{N_k} f(\omega_k^t, x_k^i) - \frac{1}{N} \sum_{k=1}^{K} \sum_{i=1}^{N_k} f(\omega_g^t, x_k^i)| \\
&= \frac{1}{N} \sum_{k=1}^{K} \sum_{i=1}^{N_k} |f(\omega_k^t, x_k^i) - f(\omega_g^t, x_k^i)| \\
&= \frac{1}{N} \sum_{k=1}^{K} \sum_{i=1}^{N_k} |f(\omega_g^{t-1} - \eta \nabla \mathcal{L}(\omega_g^{t-1}, \mathcal{D}_k), x_k^i) - f(\omega_g^{t-1} - \eta \frac{N_k}{N} \sum_{k=1}^{K} \nabla \mathcal{L}(\omega_g^{t-1}, \mathcal{D}_k), x_k^i)|.
\end{aligned}
\tag{14}
$$

It can be seen that the deviation of the mean is caused by the inconsistency between the local updates ($\nabla \mathcal{L}(\omega_g^{t-1}, \mathcal{D}_k)$) and the global updates ($\frac{N_k}{N} \sum_{k=1}^{K} \nabla \mathcal{L}(\omega_g^{t-1}, \mathcal{D}_k)$). Only when the data of each local client follows the same distribution will the local updates be consistent with the global updates, which contradicts the non-independent and identically distributed (Non-IID) in federated learning. Therefore, the greater the difference in data distribution, the greater the offset of statistical parameters.

### A.2. Obtain Unbiased Global Statistics

Recognising that statistical bias is produced during the update stage, where the local statistical parameters and learnable parameters are jointly updated, we have sufficient motivation to separate their updates. In the stage of calculating local statistics, we freeze the learnable parameters to get:

$$
\begin{aligned}
\mu_k^t &= \frac{1}{N_k} \sum_{i=1}^{N_k} f(\omega_g^{t-1}, x_k^i); \\
(\sigma^2)_k^t &= \frac{1}{N_k} \sum_{i=1}^{N_k} (f(\omega_g^{t-1}, x_k^i) - \mu_k^t)^2.
\end{aligned}
\tag{15}
$$

Then in the server stage, we can use distributed methods to aggregate these statistics to obtain an unbiased estimate of the population according to Equation (8):

$$
\begin{aligned}
\mu_g^t &= \sum_{k=1}^{K} \frac{N_k}{N} \mu_k^t \\
&= \sum_{k=1}^{K} \frac{N_k}{N} \sum_{i=1}^{N_k} \frac{1}{N_k} f(\omega_g^{t-1}, x_k^i) \\
&= \frac{1}{N} \sum_{k=1}^{K} \sum_{i=1}^{N_k} f(\omega_g^{t-1}, x_k^i) \\
&= \hat{\mu}_g^{t-1}; \\
(\sigma^2)_g^t &= \frac{N_k}{N-1} \sum_{k=1}^{K} \left[ (\sigma^2)_k^t + (\mu_k^t - \mu_g^t)^2 \right] \\
&= \frac{1}{N-1} \sum_{k=1}^{K} \left[ \sum_{i=1}^{N_k} (f(\omega_g^{t-1}, x_k^i) - \mu_k^t)^2 + N_k(\mu_k^t - \mu_g^t)^2 \right] \\
&= \frac{1}{N-1} \sum_{k=1}^{K} \left[ \sum_{i=1}^{N_k} (f^2(\omega_g^{t-1}, x_k^i) - 2f(\omega_g^{t-1}, x_k^i)\mu_k^t + (\mu_k^t)^2) + N_k((\mu_k^t)^2 - 2\mu_k^t\mu_g^t + (\mu_g^t)^2) \right] \\
&= \frac{1}{N-1} \sum_{k=1}^{K} \left[ \sum_{i=1}^{N_k} f^2(\omega_g^{t-1}, x_k^i) - \sum_{i=1}^{N_k} 2f(\omega_g^{t-1}, x_k^i)\mu_k^t + N_k(\mu_k^t)^2 + N_k(\mu_k^t)^2 - 2N_k\mu_k^t\mu_g^t + N_k(\mu_g^t)^2 \right] \\
&= \frac{1}{N-1} \sum_{k=1}^{K} \left[ \sum_{i=1}^{N_k} f^2(\omega_g^{t-1}, x_k^i) - 2N_k\mu_k^t\mu_g^t + N_k(\mu_g^t)^2 \right] \\
&= \frac{1}{N-1} \sum_{k=1}^{K} \sum_{i=1}^{N_k} (f(\omega_g^{t-1}, x_k^i) - \mu_g^t)^2 \\
&= (\hat{\sigma}^2)_g^{t-1}.
\end{aligned}
\tag{16}
$$

Therefore, we can obtain unbiased estimates of model $\omega_g^{t-1}$ on the overall data without being affected by heterogeneous data distributions.

## B. More Experiments

### B.1. Model architecture of Simple-CNN

As shown in Table 7, Simple-CNN mainly consists of three convolutional blocks and two fully connected layers. Each convolutional block contains a convolutional layer, a norm layer, a ReLU activation layer, and a max-pooling layer. We replace the norm layer with different normalisation methods, except for FedFN. For FedFN, we normalise the input before the fully connected layer according to (Zhang et al., 2024).

Table 7: Model architecture of Simple-CNN with different normalisation layers (BN/GN/LN/FixBN/FBN/FedFN/HBN).

| Simple-CNN | +BN | +GN | +LN | +FixBN | +FBN | +FedFN | +HBN |
|---|---|---|---|---|---|---|---|
| | Conv2D(3,16,3,1,1) | | | | | | |
| Block 1 | BN | GN | LN | FixBN | FBN | - | HBN |
| | ReLU | | | | | | |
| | MaxPool2D(2,2) | | | | | | |
| | Conv2D(16,32,3,1,1) | | | | | | |
| Block 2 | BN | GN | LN | FixBN | FBN | - | HBN |
| | ReLU | | | | | | |
| | MaxPool2D(2,2) | | | | | | |
| | Conv2D(32,64,3,1,1) | | | | | | |
| Block 3 | BN | GN | LN | FixBN | FBN | - | HBN |
| | ReLU | | | | | | |
| | MaxPool2D(2,2) | | | | | | |
| | - | - | - | - | - | FedFN | - |
| Block 4 | FC(1024,128) | | | | | | |
| | ReLU | | | | | | |
| Block 5 | FC(128,n) | | | | | | |

## B.2. Datasets

We mainly evaluate our algorithm on three image classification datasets:

- CIFAR-10: CIFAR-10 It consists of 60,000 32x32 color images, divided into 10 classes with 6,000 images per class.

- CIFAR-100: CIFAR-100 consists of 60,000 32x32 color images, divided into 100 classes with 6,00 images per class.

- Tiny-ImageNet: A smaller version of the ImageNet dataset. Tiny-ImageNet consists of 100,000 64x64 color images, divided into 200 classes with 5,00 images per class.

## B.3. Convergence Curves of Different Model Architectures

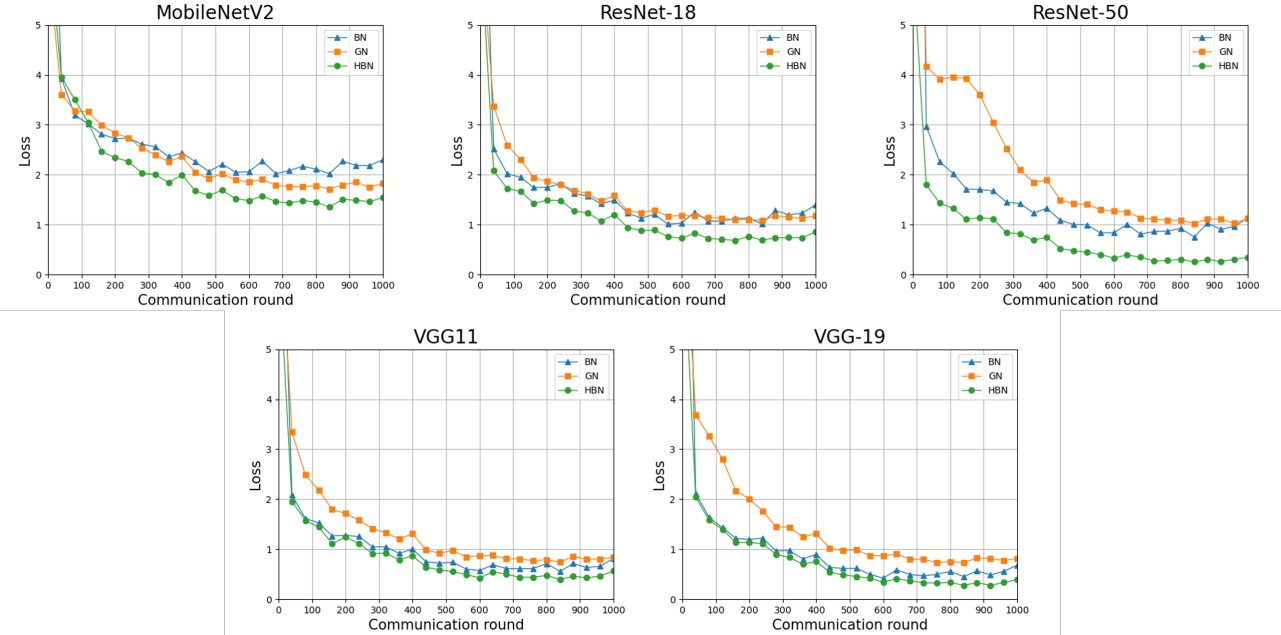

Figure 6: Convergence curves of HBN under different model architectures on CIFAR-100 with $\phi = 0.1$ and $B = 64$.

As shown in Figure 6, we plot the convergence curves of the above three normalisation methods under different model architectures, and HBN is significantly better than the other two normalisation methods in terms of convergence.

### B.4. Learning Rate Grid Search

Since different normalisation techniques operate on distinct dimensions (*e.g.*, batch, layer, or group), they induce varying scale transformations in the data. This necessitates tuning of learning rates to match their respective normalisation. We present the learning rate grid search results on three datasets in Table 8.

Table 8: Learning rate grid search results on three datasets

| CIFAR-10 ($\beta = 0.6, B = 4$) | | |
| --- | --- | --- |
| **Methods** | **Search range** | **Accuracy** |
| BN | [0.02, **0.01**, 0.005, 0.001] | [75.36, **75.83**, 74.29, 73.37] |
| GN | [0.01, **0.005**, 0.002, 0.001, 0.0005] | [74.08, **75.74**, 73.47, 73.10, 70.01] |
| LN | [0.01, **0.005**, 0.002, 0.001] | [N/A, **74.55**, 74.34, 74.0] |
| FedFN | [0.01, **0.005**, 0.002, 0.001, 0.0005] | [75.33, **75.51**, 75.49, 75.44, 74.30] |
| FixBN | [0.01, 0.005, **0.002**, 0.001] | [75.65, 76.91, **77.34**, 75.30] |
| FBN | [**0.01**, 0.005, 0.002, 0.001] | [**73.91**, 73.05, 72.44, 68.16] |
| HBN | [0.02, **0.01**, 0.005, 0.001] | [78.08, **78.22**, 77.90, 69.30] |
| CIFAR-100 ($\beta = 0.6, B = 4$) | | |
| **Methods** | **Search range** | **Accuracy** |
| BN | [0.05, 0.02, **0.01**, 0.005, 0.001] | [N/A, 45.77, **45.84**, 45.54, 45.13] |
| GN | [0.02, 0.01, 0.005, 0.002, **0.001**, 0.0005] | [N/A, 44.47, 46.70, 46.99, **47.11**, 39.25] |
| LN | [0.01, 0.005, **0.002**, 0.001, 0.0005] | [N/A, 42.43, **46.90**, 45.04, 31.48] |
| FedFN | [0.01, 0.005, 0.002, **0.001**, 0.0005] | [N/A, 43.75, 44.15, **46.29**, 35.15] |
| FixBN | [0.01, **0.005**, 0.002, 0.001, 0.0005] | [43.72, **46.05**, 43.90, 32.47, 32.48] |
| FBN | [**0.02**, 0.01, 0.005, 0.002, 0.001] | [**45.35**, 43.68, 43.33, 39.11, 35.47] |
| HBN | [0.02, **0.01**, 0.005, 0.001] | [44.93, **49.88**, 49.36, 49.11] |
| Tiny-ImageNet ($\beta = 0.1, B = 64$) | | |
| **Methods** | **Search range** | **Accuracy** |
| BN | [0.32, **0.16**, 0.08, 0.032, 0.016] | [23.45, **23.95**, 21.92, 20.07, 15.39] |
| GN | [0.32, 0.16, **0.08**, 0.032, 0.016] | [16.56, 18.43, *19.71*, 18.40, 15.71] |
| LN | [0.32, 0.16, **0.08**, 0.032, 0.016] | [16.28, 18.07, **19.72**, 18.53, 15.31] |
| FedFN | [0.32, **0.16,** 0.08, 0.032, 0.016] | [14.92, **20.05**, 19.38, 18.61, 16.17] |
| FixBN | [**0.32**, 0.16, 0.08, 0.032, 0.016] | [**24.71**, 24.62, 22.95, 20.38, 15.65] |
| FBN | [0.32, 0.16, **0.08**, 0.032, 0.016] | [N/A, N/A, **4.90**, 3.40, 2.32] |
| HBN | [**0.32**, 0.16, 0.08, 0.032, 0.016] | [**25.59**, 24.79, 23.76, 22.0, 17.21] |

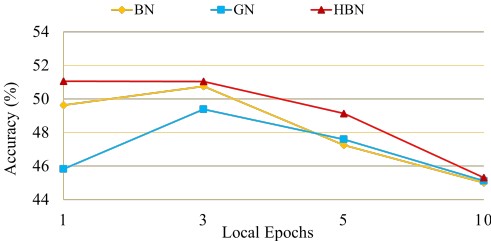

Figure 7: Experiments on CIFAR-100 ($\beta = 0.6, B = 32$) across varying local epochs by Simple-CNN.

### B.5. Different local training epochs

Figure 7 shows the changes in model accuracy with increasing local epochs for three normalisation methods. HBN maintains its advantage within an appropriate range of local epochs. When the number of epochs reaches 10, different normalisation methods no longer have a significant impact on model performance, as the primary limiting factor at this stage becomes the excessive divergence among local model parameters.

