# OpenReview forum: "Hybrid Batch Normalisation: Resolving the Dilemma of Batch Normalisation in Federated Learning"
_ICML.cc/2025/Conference — ICML 2025 poster_

### Official Review · Reviewer_b8GA · 2025-03-08

**Overall Recommendation:** 2

**Summary:**

The paper introduces Hybrid Batch Normalization (HBN) as a new normalization method designed to overcome the limitations of Batch Normalization in FL. In FL, client data is Non-IID, leading to a discrepancy between local and global statistics, which degrades BN’s performance. HBN addresses this issue by adaptively combining global and local batch statistics. The experimental results demonstrate that HBN outperforms existing normalization methods on CIFAR-10, CIFAR-100, and Tiny-ImageNet datasets.

**Claims And Evidence:**

The proposed method is sound, and the theoretical formulation aligns well with the motivation and objectives of the paper.

**Essential References Not Discussed:**

1. **Include FedBN [1] as an additional baseline**
   - FedBN is a foundational work that directly addresses **batch normalization in FL**. Adding its results would provide a more comprehensive evaluation of existing normalization methods.

2. **Cite FedFN [2] and replace FN with FedFN**
   - The paper currently references **FN**, but a more direct and relevant work that first introduced **feature normalization in FL** is **FedFN**. I recommend citing **FedFN** explicitly and renaming **FN to FedFN** for accuracy.


### **References**
[1] **FedBN** : Federated Learning on Non-IID Features via Local Batch Normalization, ICLR 2021.
[2] **FedFN**: Feature Normalization for Alleviating Data Heterogeneity Problem in Federated Learning.

**Experimental Designs Or Analyses:**

### Recommendations for Table 1

In **Table 1**, which presents experimental comparisons of different **normalization solutions in FL**, I recommend making the following modifications and additions:

1. **Include FedBN [1] as an additional baseline**
   - FedBN is a foundational work that directly addresses **batch normalization in FL**. Adding its results would provide a more comprehensive evaluation of existing normalization methods.

2. **Cite FedFN [2] and replace FN with FedFN**
   - The paper currently references **FN**, but a more direct and relevant work that first introduced **feature normalization in FL** is **FedFN**. I recommend citing **FedFN** explicitly and renaming **FN to FedFN** for accuracy.

3. **Expand the hyperparameter search space for local training epochs and learning rates**
   - **Section 4** states that the grid search for the **learning rate** was conducted over **{0.01, 0.005, 0.002, 0.001}**, with **local training epochs fixed at 1**.
   - I recommend **extending the grid search** as follows:
     - **Local training epochs**: Include **{3, 5, 10, 15}** in the search space.
     - **Learning rate**: Add **{1.0, 0.5, 0.1, 0.05}** to the search range.

   **Reasoning:**
   - As stated in **FedFN [2]**,
     > *"FedFN scales the gradient of **θ_cls** by dividing it by the feature vector norm. This scaling significantly impacts the gradient of **θ_cls** and, consequently, the applied learning rate."*
   - Similar to **FedFN**, the proposed method applies **feature normalization**, meaning it could exhibit **different optimal learning rates compared to baselines**.
   - Other FL studies that applied **scaled learning rates for feature normalization** include:
     - **SphereFed** [3]
     - **Neural Collapse Inspired FL** [4]
     - **FedDr+** [5]

   - Since the experimental setup appears to follow benchmarks similar to **[5]**, it is possible that the current **learning rate and local epoch settings are relatively small**. Expanding the **grid search range** would ensure a more robust evaluation.

4. **Provide detailed hyperparameter settings in the appendix**
   - Each **dataset and algorithm** should have a **detailed breakdown of the applied learning rate, local epoch, and other key hyperparameters** in the appendix to improve reproducibility.

---

### **References**
[1] **FedBN** : Federated Learning on Non-IID Features via Local Batch Normalization, ICLR 2021.
[2] **FedFN**: Feature Normalization for Alleviating Data Heterogeneity Problem in Federated Learning.
[3] **SphereFed**: Hyperspherical Federated Learning, ECCV 2022.
[4] **No Fear of Classifier Biases**: Neural Collapse Inspired Federated Learning with Synthetic and Fixed Classifier, ICCV 2023.
[5] **FedDr+**: Stabilizing Dot-regression with Global Feature Distillation for Federated Learning, TMLR 2025.

**Methods And Evaluation Criteria:**

here is a foundational and impactful paper, FedBN [1], that initially addressed the issue of BN in FL. The paper should introduce FedBN more explicitly and clearly highlight the differences between HBN and FedBN throughout the presentation. Additionally, further comparative experiments with FedBN would strengthen the paper’s contribution and clarify its novelty.

[1] FedBN: Federated Learning on Non-IID Features via Local Batch Normalization, ICLR 2021.

**Other Comments Or Suggestions:**

There are existing approaches **[1,2,3]** that address **data heterogeneity in FL** by **freezing the classifier and modifying the loss function**. In contrast, this study focuses on modifying the **forward pass** of the model. Given this distinction, it seems that the proposed method could be effectively combined with these prior approaches.

I am particularly interested in whether **HBN** maintains its advantage over **BN and FedBN** when integrated with the **classifier-freezing methods** from **[1,2,3]**. It would be valuable to examine whether **Simple BN vs. FedBN vs. HBN** still demonstrates performance improvements when applied alongside these approaches.

### **References**
[1] **SphereFed**: Hyperspherical Federated Learning, ECCV 2022.
[2] **No Fear of Classifier Biases**: Neural Collapse Inspired Federated Learning with Synthetic and Fixed Classifier, ICCV 2023.
[3] **FedDr+**: Stabilizing Dot-regression with Global Feature Distillation for Federated Learning, TMLR 2025.

**Other Strengths And Weaknesses:**

Apart from the points mentioned earlier, I have no additional comments.

**Questions For Authors:**

Apart from the points mentioned earlier, I have no additional comments.

**Relation To Broader Scientific Literature:**

Apart from the points mentioned earlier, I have no additional comments.

**Theoretical Claims:**

The paper does not provide any formal theoretical analysis or proofs. As a result, there are no theoretical claims to verify.

---

> ### Author Rebuttal · Authors · 2025-04-01
>
> Thank you for your valuable contributions to improving this paper. In response to your suggestions, please find our detailed replies below.
>
> **1. FedBN Baseline**
>
> FedBN is designed for personalised FL without obtaining a unified global model, as it keeps BN parameters client-specific,
> while focusing on the general FL scenarios, which train a global model for all the local clients.
> Therefore, we did not compare with FedBN in our experiments.
> However, we will include more detailed discussion of the techniques used in FedBN in our final version.
>
> **2. FedFN Terminology**
>
> We thank the reviewer for this valuable suggestion.
> We confirm that FedFN is implemented consistently with FN as described in our paper.
> To improve the accuracy of description, we will explicitly replace FN with FedFN in the revised version to avoid any potential ambiguity.
>
> **3. Hyperparameter Search**
>
> The use of inconsistent local update epochs would compromise the fairness of comparisons, as it primarily impact on the number of local updates.
> To mitigate this, we fixed the epoch value to 1 in our work.
> Similarly, the local batch size affects the communication frequency, which is why we haven't made adjustments for local update epochs.
> As you rightly observed, different algorithms exhibit distinct optimal learning rates across datasets (please refer to the anonymous link:
> https://anonymous.4open.science/r/ICML_2025-F29D/grid_search.png).
> We ensure all the methods could stably converge within the searched range. The empirical results indicate that larger learning rates prevent trainability under our small batch size constraints.
>
> **4. Combining with Classifier-Freezing Methods**
>
> In Table C, we compare the performance of the suggested three classifier-freezing approaches, integrated with our HBN, following the original experimental setup of MobileNet on CIFAR-100 (sharding partition strategies with 10 shards per client) [1].
> In our implementation, we solely replace the standard BN layers in MobileNet with HBN.
> HBN achieves encouraging performance gains.
> HBN enhances activation normalisation during training by adaptively blending local and global statistics, thereby mitigating bias introduced by relying exclusively on local statistics.
> This hybrid approach ensures more stable and representative feature distributions across heterogeneous data partitions, which is particularly critical in classifier-freezing methods.
>
> *Table C: A comparison of classifier-freezing methods with BN and HBN.*
> | Methods   | +BN   | +HBN     |
> |-----------|-------|----------|
> | FedAvg    | 34.92 | **38.86**|
> | sphereFed | 42.80 | **49.57**|
> | FedETF    | 32.30 | **45.38**|
> | FedDr+    | 47.58 | **50.22**|
>
> [1] S. Kim et al., "FedDr+: Stabilizing Dot-regression with Global Feature Distillation for Federated Learning," TMLR, 2025.

---

### Official Review · Reviewer_Dpdp · 2025-03-14

**Overall Recommendation:** 2

**Summary:**

This paper introduces Hybrid Batch Normalisation (HBN), a normalization technique designed to address the limitations of standard Batch Normalisation (BN) in federated learning (FL) with non-IID data. HBN separates the update of statistical parameters (means and variances) from learnable parameters, enabling unbiased global statistical estimates. It incorporates a learnable hybrid distribution factor to adaptively blend local batch statistics with global statistics. HBN outperforms BN, Group Normalisation (GN), in classification tasks across CIFAR-10, CIFAR-100, and Tiny-ImageNet, particularly under data heterogeneity and small batch sizes.

**Claims And Evidence:**

- The proposed two-stage update for separating statistical and learnable parameters is simple yet effective, with solid theoretical support in the Appendix.

- However, the paper does not fully explain why combining global and local statistics in local training is beneficial, and the evidence provided is limited.


- Similarly, the paper presents two conflicting claims: "Using global statistics for batch normalization in local training is helpful" versus "FixBN and FBN overlook the diversity of local client statistics, a key challenge in non-IID settings." Simply stating that prior works ignore local diversity feels unclear, as their main contribution is using shared global statistics to tackle data heterogeneity across clients. The ablation study (Table 4) shows the hybrid component drives performance gains, but there’s little analysis or theoretical backing for why adding local batch statistics improves local training.

**Essential References Not Discussed:**

Another line of research [1, 2] uses weight standardization instead of normalization to handle statistical differences between clients. I believe these studies are relevant to this work and should be included as baselines for comparison.

[1] Siomos, Vasilis, et al. "Addressing Data Heterogeneity in Federated Learning with Adaptive Normalization-Free Feature Recalibration." arXiv preprint arXiv:2410.02006 (2024).
[2] Zhuang, Weiming, and Lingjuan Lyu. "Fedwon: Triumphing multi-domain federated learning without normalization." ICLR 2024.

**Experimental Designs Or Analyses:**

- I think FedTAN should be included as a baseline for comparison.

- The main results for FixBN, FBN, and FN in Table 1 show they often perform no better than BN, which contradicts the claims and results in their original papers. If these methods are implemented correctly, the paper should discuss the reasons for this discrepancy in the main paper.

**Methods And Evaluation Criteria:**

The proposed HBN method, which separates statistical and learnable parameter updates and introduces a hybrid factor, is logically sound for addressing BN’s mismatch in FL’s non-IID settings. Evaluation on standard datasets (CIFAR-10, CIFAR-100, Tiny-ImageNet) with Dirichlet-distributed heterogeneity is appropriate and aligns with FL research norms. The use of Simple-CNN and ResNet-18 as benchmarks is reasonable, though testing on a broader range of architectures could strengthen generalizability. Metrics like top-1 accuracy are standard and suitable for the classification focus.

**Other Comments Or Suggestions:**

- The T-SNE results in Figure 2 look very similar across normalization techniques. It’s hard to tell which one is better by eye, and the differences seem extremely marginal.

**Other Strengths And Weaknesses:**

If authors provide sufficient analysis or theoretical evidence on how the hybrid normalization works, its strength would lie in effectively combining local and global statistics, offering a compelling methodology.

**Questions For Authors:**

- Is the statistics EMA technique used in the implementation also applied in other global statistics-based approaches?

**Relation To Broader Scientific Literature:**

While existing methods focus on obtaining accurate global statistics, this paper concentrates on deriving an unbiased estimator and effectively using it with local statistics. It shows that combining both is better than using global statistics alone, but it doesn’t thoroughly analyze the issues with relying solely on global statistics.

**Theoretical Claims:**

The paper provides derivations for unbiased global statistics (Appendix A.2, Equations 15-16), claiming they mitigate statistical bias in FL. I checked these proofs, and they appear mathematically correct, leveraging distributed statistical analysis to aggregate local statistics accurately. The formulation of the hybrid normalization (Equation 9) is conceptually clear, though its theoretical optimality lacks deeper justification beyond empirical success.

---

> ### Author Rebuttal · Authors · 2025-04-01
>
> We appreciate your thoughtful suggestions. Please find our response below.
>
> **1. FedTAN Baseline**
>
> FedTAN employs real-time communication to synchronise the use of shared global statistics.
> However, to obtain these global statistics, FedTAN requires three rounds of communication per BN layer during the forward propagation.
> This requirement fails to accommodate a fundamental FL constraint: communication rounds restrictions.
> While the theoretical discussion of the impact of  BN in FL provides valuable insights, we have intentionally excluded FedTAN from our comparative baselines because of these  limitations.
> We acknowledge its conceptual contributions while noting its impractical communication overhead in real-world FL applications.
> Our method adds only one round overhead compared to the original BN, making it more suitable for practical scenarios.
>
> **2. Analysis of Results**
>
> FixBN and FBN employ shared global statistics for normalisation at different stages.
> When these global statistics are reliable, they benefit local training.
> Conversely, when the global statistics are unreliable, they exhibit detrimental effects.
> Our experimental scenario in Table 1 is deliberately challenging——featuring strong heterogeneity, numerous clients, and restricted batchsize.
> In such conditions, FixBN and FBN fail to judge whether the local statistics, which are derived from diverse local models, can directly be aggregated, resulting in unreliable global statistics.
> In addition, FN is a local normalisation method that normalises features using the statistics of a single sample.
> In non-iid scenarios, FN, like GN and LN, lacks awareness of the global structure, leading to performance degradation in our setup.
>
> **3. Weight Standardisation Methods**
>
> We further analysed the weight standardisation solution, Fedwon, which achieves promising results.
> Fedwon modifies the convolution layers directly to adjust the distribution, without using the standard normalisation layer.
> Therefore, the starting point of Fedwon is different from ours.
> What is exciting is that when we combine HBN with Fedwon, as shown in Table B, Fedwon+HBN yields impressive performance improvements, demonstrating the complementary power of our design towards Fedwon.
> In our work, we want to emphasise the supportive role of BN's global statistical information in FL.
> However, inappropriate implementations of BN in FL fail to fully activate its potential.
> Our key contribution is resolving the BN's dilemma in federated learning.
> Through analysis and empirical validation, HBN proves both simple and effective.
>
> Due to the lack of available open-source code for [1], we are unable to replicate it within a limited time.
> We will provide a comprehensive analysis of weight standardisation methods in the final version.
>
> *Table B: Compatibility with weight standardisation methods.*
> | Settings| Cifar10(β=0.3)| Cifar100(β=0.3)| Tiny(β=0.1)| Tiny(β=0.05)|
> |----------------|---------------|----------------|---------------------|----------------------|
> | HBN            | 76.53         | 48.93          | 25.59               | 24.69                |
> | Fedwon         | 77.87          | 49.66          | 26.97               | 24.82                |
> | Fedwon+HBN     | **78.19**     | **50.15**      | **27.90**           | **26.69**            |
>
> [1] Siomos, Vasilis, et al. "Addressing Data Heterogeneity in Federated Learning with Adaptive Normalization-Free Feature Recalibration." arXiv, 2024.
>
> **4. Why do we need hybrid batch normalisation?**
>
> If we rely solely on local statistics, this reduces to the standard BN.
> The standard BN degrades performance in non-iid scenarios, as the local statistics are diverse among different clients.
> Consider a toy example: two clusters following Gaussian distributions, simulating data from two clients in FL.
> Normalising each cluster separately using local statistics would cause their distributions to overlap (please refer to the link:
> https://anonymous.4open.science/r/ICML_2025-F29D/adaptive_normalisation.png).
> In contrast, global normalisation preserves the global structure of two clusters.
> However, real-time global normalisation is impractical, as discussed above about FedTAN.
> Using only historical global statistics ,whose timeliness is constrained by intermittent communication, will yield a suboptimal result.
> But historical global statistics still retain valuable global structural information.
> Inspired by that, our proposed HBN adaptively combines historical global statistics with current local statistics, achieving more effective normalisation.
>
>
> **5. Amelioration of Figure 2**
>
> We replace Figure 2 with an intuitive toy example (please refer to the link:
> https://anonymous.4open.science/r/ICML_2025-F29D/adaptive_normalisation.png), where hybrid normalisation can standardise the size of two clusters while maintaining the global structure.
>
> **6. Technical Details**
>
> EMA is also used in FBN, excluding other methods.

---

### Official Review · Reviewer_jjB4 · 2025-03-15

**Overall Recommendation:** 3

**Summary:**

Due to the lack of a coherent methodology for updating BN statistical parameters, standard BN degrades the federated learning performance. This paper proposes Hybrid Batch Normalization (HBN), which separates the update of statistical parameters from learnable parameters and adaptively combines local batch statistics with global statistics. The solution aims to obtain unbiased global statistical parameters while maintaining awareness of local data distributions during training.

**Claims And Evidence:**

Strengths:

1. Important question: Since federated learning uses distributed data, how to use aggregation of local statistics in batch normalization in federated learning is an important question.

2. Adaptive hybrid approach: The learnable hybrid distribution factor that balances global and local statistics is an elegant solution that adapts to varying degrees of data heterogeneity across clients.

3. Comprehensive empirical validation: The experiments cover multiple datasets (CIFAR-10/100, Tiny-ImageNet), network architectures (Simple-CNN, ResNet-18), and varying degrees of data heterogeneity and batch sizes.

4. Compatibility with existing FL methods: The authors demonstrate that HBN can be effectively combined with various federated learning approaches (FedAvg, FedProx, FedAdam, etc.), consistently boosting performance across different methods.

Weakness:

1. It doesn't consider the scalability of client number. The paper does not present the experiment to change the number of clients for one dataset.

2. Hyperparameter sensitivity: While there is some discussion of hyperparameter selection, a more comprehensive analysis would be beneficial for practitioners, such as the selection of $\alpha, \beta and \gamma$ in eq(10)

**Essential References Not Discussed:**

Unkonw

**Experimental Designs Or Analyses:**

See Claims And Evidence part.

**Methods And Evaluation Criteria:**

See Claims And Evidence part.

**Other Comments Or Suggestions:**

None.

**Other Strengths And Weaknesses:**

None.

**Questions For Authors:**

1. In the algorithm, "// forward with gradient". Is this a typo and should be "// backforward with gradient"?

2. Could we explain the structure details of the Simple-CNN network?

3. what is the $\phi$ in the experiment?

4. What is B in the table 3?

**Relation To Broader Scientific Literature:**

Unkonw

**Theoretical Claims:**

Since sampling is out of my research area, I did not check the correctness of proof.

---

> ### Author Rebuttal · Authors · 2025-04-01
>
> Thank you for your constructive comments and suggestions.
> Please find our response below.
>
> **1. Client Number Experiment**
>
> We conducted comparative experiments on CIFAR-10 ($\beta$ = 0.6) across varying client numbers (10 clients sampled per round).
> As shown in Table A, HBN consistently outperforms baselines at all scales (from K=100 to K=1000).
>
> *Table A: Experiments on CIFAR-10 (β = 0.6) across varying client numbers.*
> | Method  | K=100 | K=200 | K=500 | K=1000 |
> |---------|-------|-------|-------|--------|
> | BN      | 75.82 | 73.88 | 68.94 | 61.25  |
> | GN      | 75.74 | 67.93 | 60.31 | 52.24  |
> | LN      | 74.08 | 69.08 | 61.22 | 51.55  |
> | FixBN   | 75.65 | 71.67 | 69.07 | 63.08  |
> | FBN     | 73.91 | 68.21 | 59.22 | 50.73  |
> | FN      | 75.51 | 70.71 | 62.02 | 53.75  |
> | **HBN(Ours)**| **78.22** | **75.76** | **72.49** | **64.95** |
>
> **2. Hyperparameter Sensitivity**
>
> We apologise for any confusion caused.
> We clarify that $\alpha$, $\beta$, and $\gamma$ are learnable parameters (not hyperparameters).
> $\alpha$ is an adaptive hybrid distribution factor, while $\beta$ and $\gamma$ are learnable affine transformations.
> In this work, they are initialised as $\alpha=0$, $\beta=0$, and $\gamma=1$ respectively.
>
> **3. Algorithm Typo**
>
> Thank you for this observation.
> What we want to express is that updating statistical parameters does not require backpropagation to calculate gradients, which can save computational costs.
> For clarity, we will modify the algorithm annotation to 'without backpropagation' and 'using backpropagation' respectively in the final version.
>
> **4. Details of Model Architecture and Symbol Explanations**
>
> Regarding the model architecture of Simple-CNN, we provide its details in Appendix B.1.
> $\phi$ is the Dirichlet distribution factor that controls the degree of label heterogeneity [1] (visualisation is provided in Appendix B.2), while $B$ is the batch size.
>
> [1] Hsu, Tzu-Ming Harry, Hang Qi, and Matthew Brown. "Measuring the effects of non-identical data distribution for federated visual classification." arXiv preprint arXiv:1909.06335 (2019).

---

### Decision · Program_Chairs · 2025-05-01

**Decision:**

Accept (poster)

**Comment:**

This paper addresses the issue of batch normalization in federated learning, and proposes hybrid batch normalization as a method that works well. Reviewers agree that this is important, and that the experiments are mostly good / wide-ranging.

The authors rebutted most comments provided by the reviewers, although reviewers did not explicitly respond to this (aside from Reviewer b8GA). There are some concerns about additional baseline methods (FedTAN and FedBN), which the authors rebutted effectively in my mind. The authors also added some results with more clients, which is good. I agree with reviewer b8GA that the authors should run experiments with more than 1 local epoch (as methods can perform much worse in this setting, and this could be biasing the effectiveness of the HBN method). This is especially the case as this method is presented as a plug-and-play improvement to existing methods: it is not clear to me that this will be an improvement when more local optimization steps, as is common in federated learning.